# Genomic Insights into Idiopathic Granulomatous Mastitis through Whole-Exome Sequencing: A Case Report of Eight Patients

**DOI:** 10.3390/ijms25169058

**Published:** 2024-08-21

**Authors:** Seeu Si Ong, Peh Joo Ho, Alexis Jiaying Khng, Benita Kiat Tee Tan, Qing Ting Tan, Ern Yu Tan, Su-Ming Tan, Thomas Choudary Putti, Swee Ho Lim, Ee Ling Serene Tang, Jingmei Li, Mikael Hartman

**Affiliations:** 1Genome Institute of Singapore (GIS), Agency for Science, Technology and Research (A*STAR), Singapore 138672, Singapore; ong_seeu_si@gis.a-star.edu.sg (S.S.O.);; 2Department of Surgery, Yong Loo Lin School of Medicine, National University of Singapore, Singapore 119228, Singapore; 3Saw Swee Hock School of Public Health, National University of Singapore, Singapore 117597, Singapore; 4Department of General Surgery, Sengkang General Hospital, Singapore 544886, Singapore; 5Department of Breast Surgery, Singapore General Hospital, Singapore 169608, Singapore; 6Division of Surgical Oncology, National Cancer Centre, Singapore 169610, Singapore; 7Breast Department, KK Women’s and Children’s Hospital, Singapore 229899, Singapore; 8Department of General Surgery, Tan Tock Seng Hospital, Singapore 308433, Singapore; 9Lee Kong Chian School of Medicine, Nanyang Technological University, Singapore 308232, Singapore; 10Institute of Molecular and Cell Biology (IMCB), Agency for Science, Technology and Research (A*STAR), Singapore 138673, Singapore; 11Division of Breast Surgery, Changi General Hospital, Singapore 529889, Singapore; 12Department of Pathology, National University Health System, Singapore 119228, Singapore; 13Department of Surgery, Woodlands Health, Singapore 737628, Singapore; 14Department of Surgery, University Surgical Cluster, National University Health System, Singapore 119228, Singapore

**Keywords:** breast pathology, breast abscess, suppurative breast lesion, tuberculous mastitis, breast cancer, mastitis, idiopathic granulomatous mastitis, somatic mutations, pathogenic mutations, whole-exome sequencing

## Abstract

Idiopathic granulomatous mastitis (IGM) is a rare condition characterised by chronic inflammation and granuloma formation in the breast. The aetiology of IGM is unclear. By focusing on the protein-coding regions of the genome, where most disease-related mutations often occur, whole-exome sequencing (WES) is a powerful approach for investigating rare and complex conditions, like IGM. We report WES results on paired blood and tissue samples from eight IGM patients. Samples were processed using standard genomic protocols. Somatic variants were called with two analytical pipelines: nf-core/sarek with *Strelka2* and GATK4 with *Mutect2*. Our WES study of eight patients did not find evidence supporting a clear genetic component. The discrepancies between variant calling algorithms, along with the considerable genetic heterogeneity observed amongst the eight IGM cases, indicate that common genetic drivers are not readily identifiable. With only three genes, *CHIT1*, *CEP170*, and *CTR9*, recurrently altering in multiple cases, the genetic basis of IGM remains uncertain. The absence of validation for somatic variants by Sanger sequencing raises further questions about the role of genetic mutations in the disease. Other potential contributors to the disease should be explored.

## 1. Introduction

Idiopathic granulomatous mastitis (IGM) poses a significant clinical challenge [1,2]. It is characterised by chronic inflammation and granuloma formation in the breast, with the precise molecular mechanisms driving its pathogenesis remaining ambiguous [3,4,5]. IGM is typically reported in women of childbearing age, who are not pregnant or lactating [6,7,8]. Unlike other forms of mastitis with known risk factors of breastfeeding, infection, or autoimmunity, IGM perplexes clinicians and researchers alike due to its elusive aetiology and diverse clinical presentation [5,9,10,11].

The rarity of IGM complicates both its diagnosis and treatment [2,12]. Specific prevalence and incidence rates are not well established in the literature due to their rarity and the diagnosis being one of exclusion [13]. The epidemiology of IGM is not well defined as precise numbers are not widely available. However, some studies do indicate that the incidence may be higher in certain geographic regions and populations in Central Asia and Southeast Asia, suggesting potential environmental or genetic predisposition [2,14]. Pathologically, IGM is characterised by lobulocentric granulomas and may be associated with fistulae, abscesses, and, in some cases, significant breast deformity [12,15,16]. The clinical course of IGM can be variable, with some patients experiencing spontaneous resolution, while others may suffer from recurrent or persistent disease requiring long-term management [12,17]. Treatment options are diverse, ranging from corticosteroids and immunosuppressants to surgical intervention, but no consensus exists on the optimal approach [2,18]. Prognosis is equally varied depending on the severity and response to treatment [12,19]. With its low incidence rate and heterogeneous clinical manifestations, establishing standardised diagnostic criteria and therapeutic guidelines proves challenging [2,8,14]. The lack of consensus regarding management also underscores the pressing need for a deeper understanding of the disease’s molecular underpinnings [13].

Advancements in genomic technologies have heralded a new era in unravelling the genetic basis of complex diseases. Whole-exome sequencing (WES) has emerged as a powerful tool for comprehensively interrogating the coding regions of the genome, offering a promising avenue to explore the genetic landscape of rare disorders like IGM [20]. Studying IGM-related somatic mutations enhances our understanding of the molecular mechanisms underlying the disease’s pathogenesis. Such insights can lead to improved diagnostics through biomarker identification, enabling quicker and more accurate differentiation from other breast diseases [21]. In this study, we embark on a first-of-its-kind endeavour to identify somatic mutations associated with IGM by employing WES on matched blood and tissue samples from IGM patients.

## 2. Results

### 2.1. Patient Demographics and Clinical Characteristics

Paired blood and breast tissue samples donated by eight women diagnosed with IGM were processed for WES. The patient demographic and clinical characteristics are shown in Table 1.

The women have a median age of 33 (interquartile range 27.3–34.5) (Table 1). The other demographic variables reported were ethnicity, body mass index, and education level (Table 1). Clinical characteristics reported were parity, number of children, smoking, chronic illness diagnosis, and first-degree family history of breast cancer (Table 1). All patients reported no alcohol consumption; no previous or existing diagnosis of autoimmune conditions (coeliac disease, type 1 diabetes mellitus, Graves’ disease, inflammatory bowel disease, multiple sclerosis, psoriasis, rheumatoid arthritis, or lupus erythematosus); and no previous or existing cancer diagnosis.

### 2.2. DNA Quality and Sequencing Metrics

Genomic DNA extraction yielded mean DNA concentration from blood samples at 12.5 ng/µL and from tissue samples at 10.7 ng/µL (Appendix A, Appendix A). WES libraries prepared had typical fragment size distributions with a peak range of 320–337 bp for the blood samples and 315–332 bp for the tissue samples (Appendix A). An average of almost 91 million reads per sample was obtained (Appendix A). The reads were aligned to the human GRCh38 reference genome using BWA, with a mean mapping rate of 100.0% (Appendix A). The average duplication rate was 17.2%, ensuring efficient use of sequencing capacity (Appendix A). The average coverage depth across target exonic regions was 37.5× (range 26.66–53.51×), ensuring high sensitivity for variant detection (Appendix A). Of the target regions, 100% were covered at least 20×, indicating uniform coverage across the exome (Appendix A). The GC content of the reads was within the expected range for human exonic sequences (range 42.3–43.7%) (Appendix A). Other summary statistics for sequencing performance, coverage metrics, and sequencing read quality control values are displayed in Appendix A.

### 2.3. Somatic Variants Identified from WES

Variant calling by *Strelka2* in nf-core/sarek pipeline and *Mutect2* in GATK4 Best Practices workflow, yielded

Variants called from blood samples: variants identified in blood samples (Appendix A);Variants called from paired blood/tissue samples: variants identified in the tissue sample that were not present in the corresponding blood sample (Table 2).

**Table 2 ijms-25-09058-t002:** Somatic variants identified from WES of paired blood/tissue samples through *Strelka2* and *Mutect2* variant calling.

Case	Somatic Variants		SNVs ^1^		Indels ^2^		PTVs ^3^		Pathogenic ^4^		Pathogenic/Likely Pathogenic ^4^		Likely Pathogenic ^4^	
	*Strelka2*	*Mutect2*	*Strelka2*	*Mutect2*	*Strelka2*	*Mutect2*	*Strelka2*	*Mutect2*	*Strelka2*	*Mutect2*	*Strelka2*	*Mutect2*	*Strelka2*	*Mutect2*
1	57	224	56	219	1	1	1	13	0	0	0	0	0	0
2	50	46	48	38	2	7	0	5	0	0	0	0	0	0
3	57	72	56	69	1	2	1	4	0	0	0	0	0	0
4	53	52	51	48	2	4	3	5	0	0	0	0	0	0
5	56	33	54	28	2	5	2	6	0	0	0	0	0	0
6	80	35	79	26	1	8	0	4	0	0	0	0	0	0
7	65	39	65	32	0	6	0	5	0	0	0	0	0	0
8	53	52	52	39	1	11	0	8	0	0	0	0	0	0
Median (range)	56.5 (50–80)	49 (23–224)	55 (48–79)	38.5 (26–219)	1 (0–2)	5.5 (1–11)	0.5 (0–3)	5 (4–13)	0 (0–0)	0 (0–0)	0 (0–0)	0 (0–0)	0 (0–0)	0 (0–0)

^1^ Single-nucleotide variants. ^2^ Insertions and deletions. ^3^ Protein-truncating variants. These correspond to variants annotated as nonsense mutations or frameshift insertions or deletions by GATK4 *Funcotator*. ^4^ *ClinVar* annotation of pathogenicity within GATK4 *Funcotator* variant annotation.

Appendix A and Table 2 show the number of somatic variants, single-nucleotide variants (SNVs), insertions and deletions (indels) called by the two variant callers for blood samples, and blood/tissue paired samples, respectively, for the eight cases. Variants annotated as “Nonsense_Mutation”, “Frame_Shift_Ins”, and “Frame_Shift_Del” with *Funcotator* were labelled protein-truncating variants (PTVs). Table 2 also shows the number of PTVs and *ClinVar* pathogenicity annotations for the eight cases.

In paired blood/tissue samples, *Strelka2* called more variants than *Mutect2* (Table 2, Figure 1a). The medians of all variants called were 56.5.5 (range 50–80) for *Strelka2* and 49 (range 23–224) for *Mutect2* (Table 2). *Strelka2* and *Mutect2* also called more SNVs than indels (Table 2, Figure 1a).

Amongst non-synonymous mutations annotated with *Funcotator*, missense mutations were annotated the most (Figure 2a).

More non-synonymous mutations and more types of non-synonymous mutations were annotated from the variants called by *Mutect2* than by *Strelka2* (Figure 2a). The median mutations annotated per sample was 49 for variants called by *Mutect2* vs. 11.5 for variants called by *Strelka2*, shown by the red-dotted line in Figure 2a. Variants called by *Strelka2* were annotated as missense mutations, splice sites, and nonsense mutations; variants called by *Mutect2* were annotated as those already mentioned as well as frameshift insertions and deletions, in-frame insertion and deletions, and non-stop mutations (Figure 2a). None of the variants called in matched blood/tissue comparisons were pathogenic or likely pathogenic, as per *ClinVar* annotation (Table 2). Fewer variants called in matched blood/tissue samples were annotated as PTVs in *Strelka2* variant calling (median 0.5, range 0–3) compared to those of *Mutect2* (median 5, range 4–13), despite the opposite comparison for the total number of variants called (Table 2, Figure 1a,b).

Further examination of PTVs from the matched blood/tissue *Strelka2* and *Mutect2* variant calling found 53 genes altered across the eight cases (Figure 3).

49 genes altered were identified from variants called by *Mutect2*, and the remaining 4 genes were identified from variants called by *Strelka2* (Figure 3). Only 3 out of the 53 genes were altered in more than 1 case (Figure 3):*CHIT1*, altered in Cases 3, 4 and 5, nonsense mutations;*CEP170*, altered in Cases 4 and 5, nonsense mutations;*CTR9*, altered in Cases 7 and 8, nonsense mutation and frameshift deletion, respectively.

The remaining genes were only altered in single cases (Figure 3).

Functional enrichment analysis of the altered genes with *enrichR* did not reveal any statistically significant (*p* < 0.05) pathways enriched (Appendix A and Appendix A). Pathways with the lowest *p*-values identified include terpenoid backbone biosynthesis (*p* = 0.0526, adjusted *p* = 0.481), protein export (*p* = 0.0549, adjusted *p* = 0.481), and protein processing in the endoplasmic reticulum (*p* = 0.0618, adjusted *p* = 0.481) (Appendix A and Appendix A). The genes *IDI2*, *SEC62*, *DNAJB12*, and *DNAH1* were implicated in these pathways (Appendix A).

A median of 2 [range 1–3] overlapping variants per sample were called by both variant callers from paired blood/tissue samples (Table 3).

All overlapping variants are single-nucleotide variants, none of which were annotated as PTVs or pathogenic or likely pathogenic (Table 3). Only one variant from Patient 4 (missense mutation) and one variant from Patient 8 (splice site) were annotated as non-synonymous mutations.

Appendix A and Figure 1b show the variants and their categorisations for the *Strelka2* and *Mutect2* variant calling in only the blood samples. Appendix A details the overlapping variants per sample called by both variant callers in only the blood samples.

### 2.4. Validation of Somatic Variants with Sanger Sequencing

A subset of variants was selected for validation using Sanger sequencing (Appendix A). None of the selected variants were validated through Sanger sequencing (Appendix A).

## 3. Discussion

This study presents somatic variants identified through WES in paired blood and breast tissue samples from eight women diagnosed with IGM. WES libraries exhibited typical fragment size distributions, high mapping rates, and sufficient coverage depth across exonic regions. Somatic variant calling revealed more variants by *Strelka2* compared to *Mutect2* in paired samples. This flips when variants called by *Mutect2* are annotated to more non-synonymous mutations and PTVs than those called by *Strelka2*. However, none of the variants were pathogenic per *ClinVar* annotation. Further examination identified 53 altered genes, with the *CHIT1*, *CEP170,* and *CTR9* genes altered in more than one case. Functional enrichment analysis did not show statistically significant pathways, although terpenoid backbone biosynthesis, protein export, and protein processing in the endoplasmic reticulum were implicated. Validation of variants through Sanger sequencing did not yield any validated variants.

Differences in the variability of sensitivity and specificity between different variant-calling algorithms have been extensively discussed [22,23,24,25,26]. Their differences in the algorithm and focus of the variant caller underscores this discrepancy between *Strelka2* and *Mutect2* in terms of the number of variants identified and their limited overlap. *Strelka2* uses a probabilistic model leveraging local assembly and realignment to call variants for more sensitivity in identifying low-frequency somatic mutations, especially in matched tumour–normal pairs, by using Bayesian methods to model both the tumour and normal samples [27]. Contrastingly, the haplotype-based approach in *Mutect2* employs a sophisticated filtering process that incorporates various sources of evidence to distinguish between true mutations and sequencing artefacts [28]. Designed to balance sensitivity and specificity, *Mutect2* minimises false positives by implementing additional artefact filters for oxidative artefacts and strand bias, on top of the standard filtering preprocessing [28]. Furthermore, nf-core/sarek *Strelka2*’s use of hard filters based on fixed thresholds vs. GATK4 *Mutect2*’s use of machine learning to filter variants could provide an additional explanation for the large discrepancy in called variants [27,28,29].

Both algorithms identified more SNVs than indels, which is consistent with typical findings in WES studies [30,31,32]. Despite the larger number of variants identified by *Strelka2*, the fewer non-synonymous mutations and PTV annotation in *Funcotator* contrasted with *Mutect2* variant calls in matched blood/tissue samples emphasises the need for the multiple-variant-caller approach for capturing the full spectrum of genetic alterations [33,34]. The limited overlap across the different categorisations of the variants in both the matched blood/tissue calls and the blood-only calls signals that further optimisation of the variant-calling pipelines and validating identified variants through independent methods are necessary.

From the PTVs from the matched blood/tissue *Strelka2* and *Mutect2* variant calling, it was observed that *CHIT1*, *CEP170*, and *CTR9* were altered in more than one case. *CHIT1* has been implicated in both granulomatous and non-granulomatous inflammatory conditions, including multiple sclerosis, sarcoidosis, inflammatory bowel disease, and a few fibrotic interstitial lung diseases (tuberculosis, idiopathic pulmonary fibrosis, scleroderma-associated interstitial lung diseases, and chronic obstructive pulmonary diseases) [35,36,37,38,39]. Song and Shao (2024) have also proposed *CHIT1* as 1 of 12 genes in an immune-mediated genetic prognostic risk score model when administering immunotherapy in triple negative breast cancer [40]. *CEP170* and *CTR9* are involved in cell cycle processes, in which dysregulation has been described to result in the secretion of inflammatory factors, impair immune-mediated processes, and increase inflammation [41,42,43,44,45]. Potentially, the alterations in *CHIT1*, *CEP170*, and *CTR9* may individually or collectively contribute to granuloma formation and chronic inflammation in the breast tissue in IGM.

Unfortunately, there is considerable genetic heterogeneity amongst the eight IGM cases since the remaining 50 genes are each altered in single cases. This variability complicates efforts to pinpoint common genetic drivers of the disease and could suggest IGM may arise from multiple genetic pathways. Such heterogeneity is consistent with the clinical diversity observed in IGM, where patients present with a wide range of symptoms and disease severities [4,13,46]. However, the lack of statistically significant pathways identified from functional enrichment analysis of the 53 genes, with all pathways identified enriched by only one to two genes, suggests there may not be identifiable genetic drivers for IGM among these eight patients.

Both pipelines rely on rigorous variant calling and annotation processes to maximise the reliability and validity of the identified somatic variants. However, the discrepancies observed between these two pipelines identifies a significant limitation: the potential variability introduced by different analytical methods. While employing multiple pipelines can enhance the robustness of variant detection, it also raises concerns that some of the identified variants may be artifacts unique to the algorithms rather than true genetic mutations [47,48]. Standardised variant-calling practices need to be specified for studies investigating rare diseases like IGM, where the small sample size can exacerbate the effect of such analytical discrepancies [49,50,51,52]. This variability cautions interpretation of the results and demands validation of detected variants to confirm their authenticity.

Unfortunately, none of the selected variants identified from WES were validated with Sanger sequencing. Despite achieving high coverage depth across all exome-targeted regions, WES is prone to inaccuracies, due to sequencing artefacts or accurately identifying variants in regions of genomic instability [53]. False positives can arise in short-read technology, particularly in regions with high GC content or repetitive sequences [54]. Sanger sequencing, with its ability to provide uniform coverage and longer read lengths, is a valuable orthogonal validation tool [55]. Other studies have also described somatic variants identified from WES that were not found in Sanger sequencing [31,32,56,57]. Discrepancies between WES and Sanger sequencing results can be attributed to their inherent differences in their error profiles or limitations in detecting variants present at low allele frequencies, especially in heterogeneous samples like those from IGM patients [58,59,60].

It must be recognised that the application of WES to the study of IGM presents its own unique set of challenges. WES mainly targets the exonic regions of the genome and is not as effective in identifying large structural variations, including deletions, duplications, inversions, and translocations [61]. Additionally, the rarity of IGM limits access to large patient cohorts. Not only does this inherently restrict the generalisability of the findings from this study with a sample size of eight patients, but potential genetic patterns that could be more apparent in a larger cohort may also be obscured. A larger cohort is needed for comprehensive genomic analysis with sufficient power to detect smaller effect sizes and lower impact somatic mutations [62]. Given our sample size of eight patients with matched blood and tissue samples, this study has low statistical power (0.230) to detect a significant difference in somatic mutations between the paired samples if such a difference truly exists [63]. The ideal power of at least 0.80 requires an odds ratio of approximately 7.25 to detect a statistically significant difference in somatic mutations between the paired blood and tissue samples [63]. Genetic variations relevant to IGM, with smaller effect sizes, might have been overlooked, highlighting the need for future studies with larger cohorts to validate and expand upon these findings.

Another difficulty lies in the disease’s inherent heterogeneity [2,4,11,13]. With a broad spectrum of clinical features, ranging from localised breast masses to diffuse inflammatory changes, identifying consistent genetic signatures associated with IGM is challenging [46]. Moreover, the multifactorial and inflammatory nature of IGM adds another layer of complexity to the study of its aetiology [11,64]. While this study examines the genetic factors, it is important to recognise that IGM likely results from a combination of genetic and non-genetic influences [12,65,66]. Various hypotheses, including immune dysregulation, infectious triggers, and hormonal influences, have been proposed, but the precise interplay of genetic and environmental factors remains poorly understood [11,14]. Understanding these complex interactions is key to determining the underlying causes of IGM. A multifaceted approach incorporating genetic studies with investigations into environmental factors, immunological responses, and hormonal profiles is needed to explore these interactions in a comprehensive manner.

This study pioneers the investigation into somatic variants in IGM patients. While the matched blood/tissue WES variant calls did not identify any *ClinVar* annotated pathogenic variants, the detection of variants in multiple genes suggests that IGM may involve a variety of molecular mechanisms. Larger studies with more comprehensive datasets are needed to uncover significant genomic drivers and biological pathways associated with IGM. Furthermore, larger scale studies could also unearth possible associations between different variants and the clinical manifestations and severity of IGM [2,13]. Future studies should also integrate additional omics data, such as transcriptomics, proteomics, and epigenomics. This will broaden the scope of research across both genetic and non-genetic factors by exploring gene expression changes, protein-level modifications and interactions, and regulatory mechanisms like DNA methylation and histone modifications [67]. Capturing data from the various biological layers will provide a more comprehensive understanding of how the genetic, environmental, hormonal, immune-mediated, and other potential uncovered factors converge to influence IGM’s pathogenesis. This comprehensive perspective will clarify the interplay between different molecular pathways involved in the disease.

The challenges in validating somatic variants underscore the need for improved methodologies and protocols for variant validation. Addressing these challenges requires a multifaceted approach, including refining bioinformatics pipelines to mitigate false positives, and an integrated orthogonal validation approach to ensure the accuracy of variant calls.

## 4. Materials and Methods

### 4.1. Patient Recruitment

The study population and patient recruitment have been previously described [11]. In brief, adult female patients with IGM were recruited from five participating hospitals in Singapore between 2018 and 2020. IGM diagnoses were based on breast core biopsy histopathology for non-caseating granulomatous inflammation and absence of malignancy. Patients were also negative for Mycobacterium tuberculosis infection (acid-fast bacillus stain) and fungal infection (Grocott’s (methenamine) silver stain or periodic acid–Schiff stain). Study coordinators sought written informed consent from potential IGM patients identified by clinicians and physicians. All studies were performed in accordance with the Declaration of Helsinki. This study was approved by the National Healthcare Group Domain Specific Review Board (reference number: 2017/01057) and the Agency for Science, Technology and Research Institutional Review Board (reference number: 2020–152).

### 4.2. Sample Collection

A subset of eight IGM patients who donated paired blood and core tissue biopsy samples for research use were included in our study (Figure 4).

Fresh blood samples were collected with DNA/RNA Shield Blood Collection tubes (catalogue number R1150; Zymo Research, Irvine, CA, USA). Fresh tissue samples were obtained with ultrasound-guided core tissue biopsies from three different regions of the affected breast area. Tissue cores were collected in 2 mL collection tubes with 300 µL DNA/RNA Shield without breads (catalogue number R1100-250; Zymo Research, Irvine, CA, USA).

### 4.3. DNA Extraction and Sequencing

Genomic DNA was extracted from the collected whole blood and fresh tissue samples, and the WES library was prepared with standard protocols. Briefly, DNA extraction was performed with *Quick*-DNA Miniprep Plus Kit (catalogue number D4069; Zymo Research, Irvine, CA, USA) according to the manufacturer’s instructions. The WES library was prepared with NEBNext^®^ Ultra™ II DNA Library Prep Modules for Illumina^®^ (catalogue number E7645L; New England Biolabs, Ipswich, MA, USA) according to the manufacturer’s instructions. Exome capture was performed with NimbleGen SeqCap EZ Exome Library Kit v3.0 (catalogue number 06465692001; Roche, Basel, Switzerland). DNA concentrations and quality were measured after extraction, shearing, pre-exome capture, and post-exome capture for quality control. Libraries were sequenced with 2 × 150 bp paired-end reads on HiSeq4000.

### 4.4. Quality Control and Somatic Variant Calling

Two analytical pipelines were used to identify somatic variants in the paired blood and tissue samples. The nf-core/sarek pipeline (version 3.3.0) was executed for identifying somatic mutations with singularity [68,69,70,71,72,73]. Raw sequencing reads were aligned to human GRCh38 reference genome (version from 22 July 2016, Broad Institute, from https://console.cloud.google.com/storage/browser/genomics-public-data/resources/broad/hg38/v0;tab=objects?prefix=&forceOnObjectsSortingFiltering=false, accessed on 19 October 2023) with BWA [74]. GATK4 was applied according to GATK Best Practices recommendations parameters for hard filtering and score recalibration for removing duplicates and base quality score recalibration [28,75,76]. *Strelka2* was used for matched tissue–normal pair variant calling [27].

Another analytical pipeline, also applying GATK4 according to the GATK Best Practices workflow, was also used to identify somatic mutations [28,75,76]. Preprocessing sequencing reads followed the same parameters as above, but the different processes were applied individually in the same sequence, without the wrapped container. Briefly, the raw sequencing reads were also aligned to the human GRCh38 reference genome using BWA [74]. Picard tools were utilised to mark and remove duplicates [28], and GATK4 *BaseRecalibrator* and GATK4 *ApplyBQSR* were utilised for base quality score recalibration, consistent with GATK Best Practices [28,75,76]. Matched tissue–normal pair somatic variant calling was executed using GATK4 *Mutect2* [28]. The called variants underwent filtering with GATK *FilterMutectCalls* before variant annotation [28].

Variants called with *Strelka2* in the nf-core/sarek pipeline and variants called with *Mutect2* in the GATK Best Practices workflow were annotated with GATK4 *Funcotator* [28]. Variant annotation was performed against *Funcotator* data sources v1.7.20200521s (GRCh38), encompassing the following databases:Catalogue of Somatic Mutations in Cancer (COSMIC) Cancer Gene Census (CGC) (data source dated 15 March 2012) [77];National Center for Biotechnology Information (NCBI) ClinVar (data source dated 29 April 2018) [78,79]; also used to perform clinical pathogenicity annotation;NCBI dbSNP (data source dated 18 April 2018) [80];Human DNA repair genes (data source dated 24 May 2018) [81];The Familial Cancer Database (FaCD) [82,83];GENCODE (v34) [84];Genome Aggregation Database (gnomAD) (v3.1.2) [85];Human Genome Organisation (HUGO) Gene Nomenclature Committee (HGNC) Database (data source dated 30 November 2017) [86].

### 4.5. Validating Called Variants with Sanger Sequencing

A subset of somatic variants identified through WES and visualised with Integrative Genomics Viewer (IGV, version 2.17.0) was validated with Sanger sequencing [87]. Primers for PCR amplification of the regions containing the variants were designed using Primer3web (version 4.1.0) [88,89,90]. The PCR products were purified using AMPure XP beads (product number A63882; Beckman Coulter Life Sciences, Indianapolis, IN, USA) in a ratio of 1:1 sample-to-bead volume. Purified PCR products were Sanger-sequenced. The sequencing data were analysed using the CLC Main Workbench (version 7.7.3, QIAGEN, Hilden, Germany) to confirm the presence or absence of the variants. The validity of somatic variants identified was determined based on sequence quality, allele frequency, and variant presence in the corresponding matched blood samples.

### 4.6. Gene Set Analysis

Functional enrichment analysis of gene sets of somatic variants identified was performed with *enrichR* (version 3.2) using the 2019 version of the Kyoto Encyclopedia of Genes and Genomes (KEGG) knowledge base [91,92,93,94,95]; comprehensive analysis and visualisation of the somatic variants identified were performed with *maftools* (version 2.6.05) [95]; and other data visualisation was performed with *ggplot2* (version 3.4.4) [96]. All analyses and visualisations were performed in *R* (version 4.0.4) unless otherwise stated.

### 4.7. Data and Sample Availability Statement

The datasets used and analysed in the current study are available from the corresponding author on reasonable request, within limitations of this study’s Institutional Review Board (IRB).

## 5. Conclusions

The absence of clear genetic drivers suggests that IGM may be influenced by non-genetic factors. Other potential contributors to the disease should be explored.

## Figures and Tables

**Figure 1 ijms-25-09058-f001:**
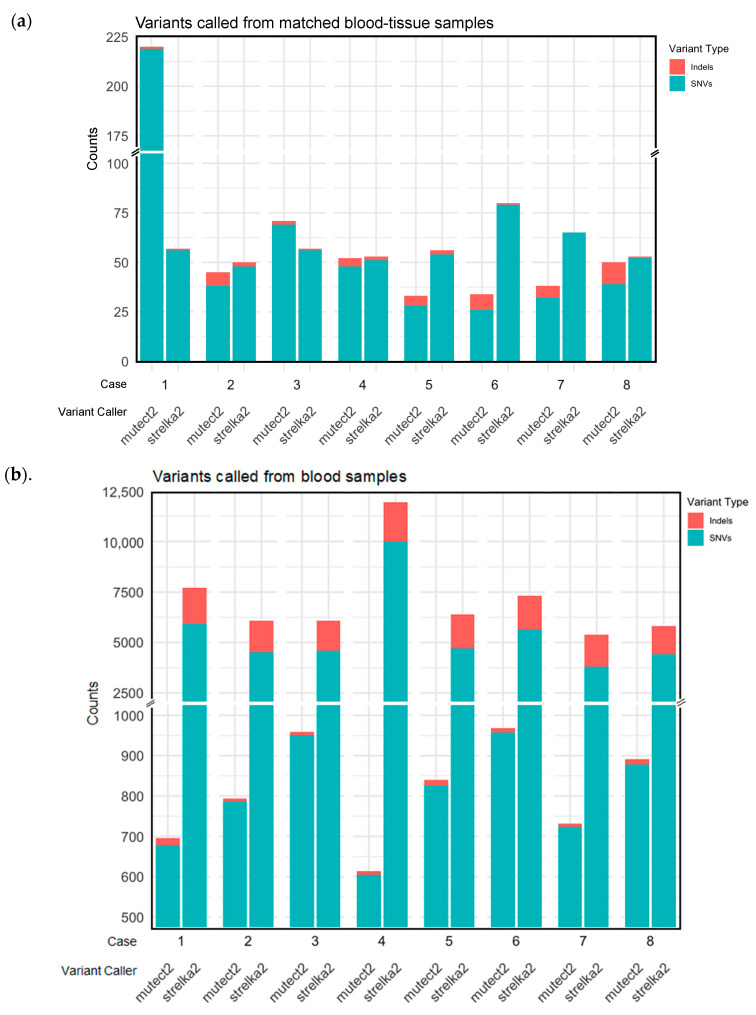
Single-nucleotide variants (SNVs) and insertion and deletion variants (indels) called by *Strelka2* and *Mutect2*, from (**a**) matched blood/tissue samples and (**b**) blood samples.

**Figure 2 ijms-25-09058-f002:**
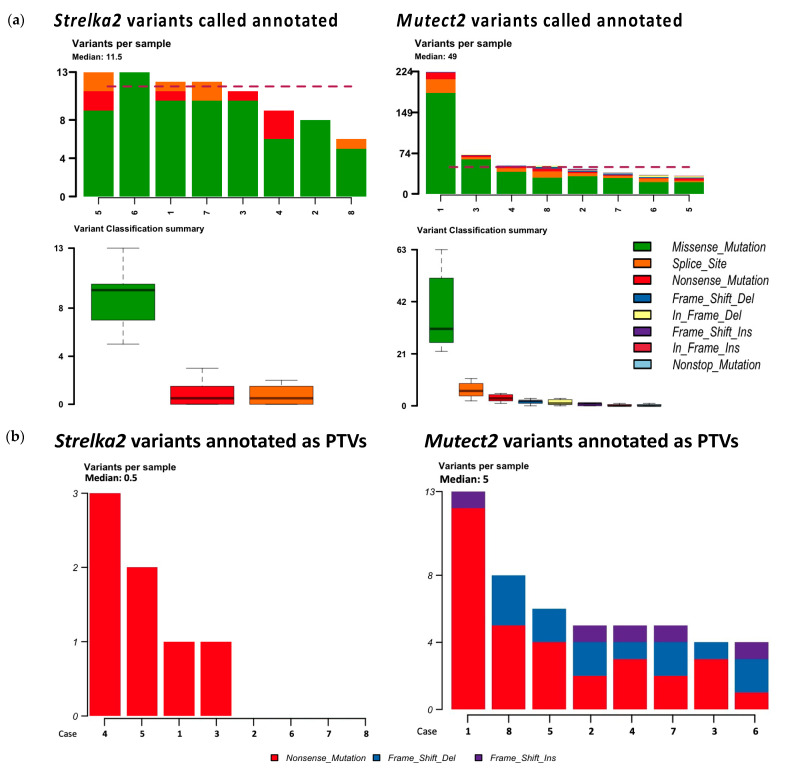
Mutations annotated by *Funcotator* from variants called with *Strelka2* and *Mutect2* matched blood/tissue samples. (**a**) All non-synonymous mutations annotated from *Strelka2* and *Mutect2* variant calls. Red-dotted line represents median of 11.5 and 49 variants per sample annotated, respectively. (**b**) Protein-truncating variants (nonsense mutations and frameshift insertions and deletions) annotated from *Strelka2* and *Mutect 2* variant calls.

**Figure 3 ijms-25-09058-f003:**
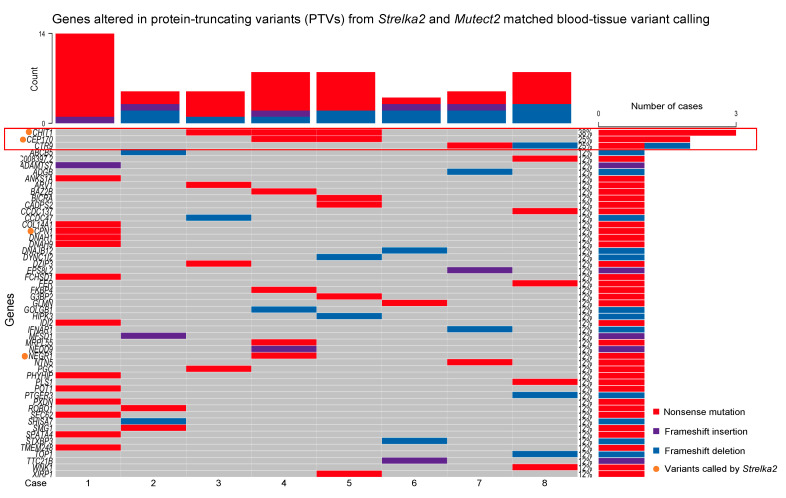
Altered genes in PTVs from *Strelka2* and *Mutect2* matched blood/tissue variant calling.

**Figure 4 ijms-25-09058-f004:**
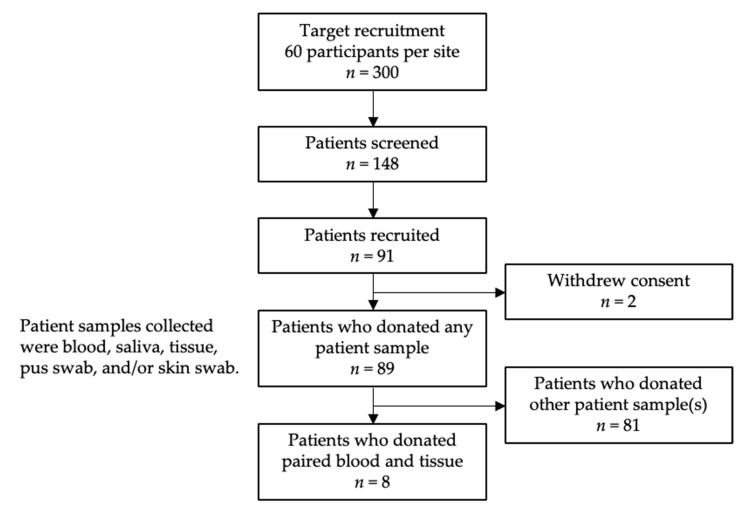
Patient recruitment, inclusion, and exclusion flowchart.

**Table 1 ijms-25-09058-t001:** Description of demographic and clinical parameters of idiopathic granulomatous mastitis (IGM) patients.

Demographic and Clinical Parameters	IGM *n* = 8
*Demographics*	
Median age at diagnosis (years, IQR)	33.0 (27.3–34.5)
Ethnicity (n, %)	
Chinese	4 (50)
Malay	3 (38)
Others	1 (12)
Body mass index (kg/m^2^, IQR)	27.770 (23.460–33.454)
Education level (n, %)	
Up to secondary school	3 (38)
Secondary school to pre-university	3 (38)
Tertiary education	2 (25)
*Patient characteristics*	
Parity (n, %)	
Yes	6 (75)
No	2 (25)
Number of children (n, %)	
No children	2 (25)
1–2 children	5 (62)
More than 2 children	1 (12)
Smoking (n, %)	
Yes	2 (25)
No	6 (75)
Chronic illness ^1^ diagnosis (n, %)	
Yes	2 (25)
No	6 (75)
Family history ^2^ of breast cancer (n, %)	
Yes	1 (12)
No	7 (88)

^1^ Chronic illness: Heart attack, stroke, or high blood pressure. ^2^ First-degree family history: Breast cancer in a parent, sibling, or child.

**Table 3 ijms-25-09058-t003:** Somatic variants identified from WES of paired blood/tissue samples that overlap in *Strelka2* and *Mutect2* variant calling.

Case	Somatic Variants	SNVs ^1^	Indels ^2^	PTVs ^3^	Pathogenic ^4^	Pathogenic/Likely Pathogenic^4^	Likely Pathogenic^4^
1	3	3	0	0	0	0	0
2	2	2	0	0	0	0	0
3	1	1	0	0	0	0	0
4	1	1	0	0	0	0	0
5	2	2	0	0	0	0	0
6	2	2	0	0	0	0	0
7	3	3	0	0	0	0	0
8	2	2	0	0	0	0	0
Median (range)	2 (1–3)	2 (1–3)	0 (0–0)	0 (0–0)	0 (0–0)	0 (0–0)	0 (0–0)

^1^ Single-nucleotide variants. ^2^ Insertions and deletions. ^3^ Protein-truncating variants. These correspond to variants annotated as nonsense mutations or frameshift insertions or deletions by GATK4 *Funcotator*. ^4^ *ClinVar* annotation of pathogenicity within GATK4 *Funcotator* variant annotation.

## Data Availability

The datasets used and analysed in the current study are available from the corresponding author on reasonable request, within the limitations of this study’s Institutional Review Board (IRB).

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
