# Peer review of "Genomic Insights into Idiopathic Granulomatous Mastitis through Whole-Exome Sequencing: A Case Report of Eight Patients"

_ijms, 2024, doi:10.3390/ijms25169058_

Round 1

Reviewer 1 Report

Comments and Suggestions for Authors

Idiopathic granulomatous mastitis (IGM) is a rare breast condition marked by chronic inflammation and granuloma formation, with an unclear cause. In this study, whole-exome sequencing (WES) was conducted on blood and tissue samples from eight IGM patients, focusing on protein-coding regions where disease-related mutations often occur. Despite using two analytical pipelines (nf-core/sarek with Strelka2 and GATK4 with Mutect2), no consistent genetic component was found. Only three genes—CHIT1, CEP170, and CTR9—showed recurrent alterations, suggesting IGM might not be primarily genetic.

1. In the abstract, the conclusion seems not cautious. The authors concluded that their study underscores the importance of considering environmental or immune-related factors in the etiology of IGM. However, this raises the question: why focus solely on environmental or immune-related factors? Are there other potential contributors that should be explored?
2. In the conclusion, the authors said Despite these obstacles, this research is a crucial advance in decoding the genetic landscape of IGM and they recommended large-scale investigations, rigorous validation studies, and the integration of multi-omics approaches. But if they concluded that no consistent genetic component was found. What are the large-scale investigations, rigorous validation studies, and the integration of multi-omics approaches? Can we be more specific?
3. Supplementary Materials are abundant.

Reviewer 2 Report

Comments and Suggestions for Authors

These authors collected and analyzed 8 cases of idiopathic granulomatous mastitis (IGM) through whole exome sequencing, which is a pilot study that has not been reported before. Although three recurrent mutations were noted, they could not be confirmed by Sanger sequencing. As a result, no significant mutation has been found. This reviewer thinks that the hypothesis that IGM may be caused by gene mutation may be off the mark because the disease is not a tumor but an inflammation. Anyway, I do not object to report this study because there are no problems with the research process.

I make two comments below:

1.      In the introduction, a brief summary regarding epidemiology, pathology, treatment, prognosis and so on of IGM should be stated, because it is a very rare disease unfamiliar with most audience.

2.      The content of conclusion is mostly meaningless. In the conclusion, it should be briefly stated that IGM may be caused by non-genetic factors.

Reviewer 3 Report

Comments and Suggestions for Authors

The article presents a study using whole-exome sequencing (WES) to investigate the genetic basis of idiopathic granulomatous mastitis (IGM). While the study is commendable for its use of advanced genomic techniques and its attempt to uncover the potential genetic background of a rare and complex disease, several concerns stand out in the analysis.

Comments

A significant limitation of the study is its small sample size (only eight patients). While this is understandable, given the rarity of IGM, the limited number of samples restricts the generalizability of the findings and may obscure potential genetic patterns that could be more apparent in a larger cohort. Furthermore, the discrepancies observed between the results of the two variant-calling algorithms indicate potential variability in the data analysis process. While the use of multiple pipelines is a strength, the lack of consistent findings between them could suggest that some of the detected variants may be artifacts of the specific algorithms used rather than true genetic mutations.

To deal with the concerns, the authors should discuss the limitations in more detail.

The study makes a valuable contribution to the understanding of IGM by employing cutting-edge genomic techniques to explore its potential genetic underpinnings. However, the small sample size, lack of variant validation, and inconsistent results between different analytical tools highlight significant challenges in this type of research. While the study raises important questions about the role of genetic factors in IGM, its findings suggest that a broader approach, considering both genetic and non-genetic factors, may be necessary to fully understand and address this rare condition. 
